# A Study on the Influence of Sensors in Frequency and Time Domains on Context Recognition

**DOI:** 10.3390/s23125756

**Published:** 2023-06-20

**Authors:** Pedro de Souza, Diógenes Silva, Isabella de Andrade, Júlia Dias, João Paulo Lima, Veronica Teichrieb, Jonysberg P. Quintino, Fabio Q. B. da Silva, Andre L. M. Santos

**Affiliations:** 1Centro de Informática, Universidade Federal de Pernambuco, Recife 50740-560, PE, Brazil; dwfs2@cin.ufpe.br (D.S.); isfa@cin.ufpe.br (I.d.A.); jdts@cin.ufpe.br (J.D.); jpsml@cin.ufpe.br (J.P.L.); vt@cin.ufpe.br (V.T.); fabio@cin.ufpe.br (F.Q.B.d.S.); alms@cin.ufpe.br (A.L.M.S.); 2Visual Computing Lab, Departamento de Computação, Universidade Federal Rural de Pernambuco, Recife 52171-900, PE, Brazil; 3Projeto CIn-UFPE Samsung, Centro de Informática, Av. Jorn. Anibal Fernandes, s/n, Recife 50740-560, PE, Brazil; jpq@cin.ufpe.br

**Keywords:** adaptive AI, context recognition, sensor fusion

## Abstract

Adaptive AI for context and activity recognition remains a relatively unexplored field due to difficulty in collecting sufficient information to develop supervised models. Additionally, building a dataset for human context activities “in the wild” demands time and human resources, which explains the lack of public datasets available. Some of the available datasets for activity recognition were collected using wearable sensors, since they are less invasive than images and precisely capture a user’s movements in time series. However, frequency series contain more information about sensors’ signals. In this paper, we investigate the use of feature engineering to improve the performance of a Deep Learning model. Thus, we propose using Fast Fourier Transform algorithms to extract features from frequency series instead of time series. We evaluated our approach on the ExtraSensory and WISDM datasets. The results show that using Fast Fourier Transform algorithms to extract features performed better than using statistics measures to extract features from temporal series. Additionally, we examined the impact of individual sensors on identifying specific labels and proved that incorporating more sensors enhances the model’s effectiveness. On the ExtraSensory dataset, the use of frequency features outperformed that of time-domain features by 8.9 p.p., 0.2 p.p., 39.5 p.p., and 0.4 p.p. in Standing, Sitting, Lying Down, and Walking activities, respectively, and on the WISDM dataset, the model performance improved by 1.7 p.p., just by using feature engineering.

## 1. Introduction

Frequent physical activity can prevent diseases in the long term, such as heart failure, diabetes, cholelithiasis, and chronic bronchitis [1]. Researchers and healthcare professionals usually measure patients’ physical activity with self-report questionnaires, which heavily rely on patients’ memory, making it a burdensome and not reliable task [2,3].

A technology that automatically monitors a patient’s daily routine and provides information to supervise their lifestyle using wearable devices can help people to achieve lifestyle changes. Smartwatches are adequate for this goal, as they do not need to be held or carried; they can comfortably stay with the user for much time.

In the early 1980s, Seiko launched the first smartwatch, which had a keyboard for data entry and could be plugged into computers to transfer data [4]. Since then, smartwatches have evolved and encapsulated many sensors. Those sensors measure motion, temperature, heart rate, electrocardiogram, and so forth [5].

Motion information can usually be acquired using three sensors: accelerometers, gyroscopes, and magnetometers. Accelerometers measure linear acceleration in space; gyroscopes provide the rotation angles; and magnetometers provide the north direction relative to the device’s local reference [6]. Understanding the movement of a person is helpful in Human Activity Recognition (HAR), so recent approaches use smartphone and smartwatch accelerometers [7,8].

Besides motion sensing, there are sensors that can provide context information, such as geographical location, microphone, and phone state. Therefore, sensor fusion approaches have been used in activity recognition to include context information. Some sensors can perform better in recognizing a specific activity than others [9]. Thus, analyzing sensor data to determine the best combination of features is vital in terms of context prediction.

Each activity has ideal duration, frequency, and intensity. Therefore, identifying the activity is essential to providing accurate recommendations such as speeding, adjusting breathing, or controlling maximum heart rate to keep the exercise rhythm moderate [10].

Complex activities are interrelated, so we can use contextual information to precisely understand a person’s lifestyle, e.g., lying on the couch is completely different from lying on a bed to sleep. For instance, if we recognize when the activity is sleeping, we can count how many hours the person sleeps daily to verify if they are sleeping enough.

Machine Learning (ML) algorithms are used to classify activities and their contexts, using as input trivial features extracted from sensor time series, such as mean, variance, and standard deviation [11]. However, although those features are easy to calculate, they may lack important information about sensors’ signals.

On the other hand, Fast Fourier Transform (FFT) algorithms may have more relevant information about sensors’ signals when they carry undesirable components such as movement artifacts and electromagnetic noise. When a signal is expressed in the frequency domain, we can pre-process the signal. Pre-processing might include filtering and feature extraction before training a model for Context Human Activity Recognition (CHAR) [12,13].

This work is a comparative study on features directly extracted from time series using trivial statistics features and features extracted using the signal Fourier Transform. We used a model proposed by [14], which hierarchically classifies complex activities. The experiments were performed on two CHAR datasets: ExtraSensory and WISDM. We also conducted a complete study of the number and type of sensors to evaluate performance in recognizing specific activities.

The results show that features extracted from the FFT series outperform the features extracted using trivial statistics. In addition, we determined that some sensors contribute more to a specific activity than others. Finally, we proved that using the sensor fusion approach can improve the model’s performance in the CHAR task.

The contributions of the present work are as follows:
1Generation of a set of features using the frequency domain and the analysis of the most relevant features for the CHAR task;2Quantitative comparison of the results using time-domain and frequency-domain features for context recognition;3Study on the contribution of every sensor and combination of sensors to recognizing specific activities and contexts.

## 2. Related Works

Neural Networks and Deep Learning algorithms are used in many different applications in the healthcare area, such as gene classification [15], novel image diagnostic methods [16], and in support of medical decisions [17]. The CHAR system can be considered a long-term-effect healthcare system, as it aims to change a person’s routine by making suggestions of new activities to the user.

HAR and CHAR have become the subject of study of many works. While HAR concentrates on the activity itself, CHAR tries to infer the context in which an action is performed. In addition, many different sensors and devices are used for both tasks, such as smartphones and smartwatches.

For HAR using context dependency, many approaches have been proposed. For example, Garcia et al. [18] combined individual and community data to train a more personalized model to recognize context. This combination was used to compensate for peculiar characteristics that may cause miss-inference due to people’s different ways of performing an ordinary activity.

Wearable sensors are the most common inputs utilized for HAR because they constantly monitor a person’s behavior. For example, Vaizman et al. [9,19] created a mobile application for HAR using wearable sensors such as accelerometers, gyroscopes, and magnetometers. Then, they collected data from 60 participants to compile the ExtraSensory dataset. In addition, they used a Neural Network for the HAR task.

Ge and Agu [20] developed QCRUFT (Quarternation Recognition under Uncertainty using Fusion and Temporal Learning). This end-to-end framework aims to recognize human activity “in the wild” and corrects noisy context labels, phone placement, and sensors’ variability. Fazli et al. [14] used a hierarchical ensemble of Multilayer Perceptrons (MLPs) to classify activities into two more general hierarchy classes.

Asim et al. [11] used the accelerometer data from ExtraSensory and feature selection techniques to propose a model for CHAR focusing on 6 primary labels and 15 secondary labels. They evaluated model performance using six ML algorithms: Random Forest, Decision Tree, Bagging, K-Nearest Neighbor, Support Vector Machine, and Naive Bayes. The same approach was also used in [7,21,22].

Another approach to CHAR is to use a multimodal platform of sensors in combination with more complex classifiers, as in [23,24,25]. Newek et al. [26] used a Random Forest classifier and the fusion of multi-sensor platforms for mobile and wearable sensors for CHAR, obtaining the accuracy of 94.23%.

Others utilized more complex classifiers to evaluate sensor fusion arrangements. A Convolutional Neural Network (CNN) was used for the CHAR task in [27]. In [28], the authors proposed a novel network model to work with highly dimensional data and compared this model against a classical Neural Network (NN).

The most critical steps in modeling a system that uses NN or ML algorithms are data analysis and feature extraction, since they can lead to miss-inference or even make the training of the model difficult due to the complexity of the problem. In [29], a Genetic Algorithm was developed with the objective of deep feature extraction to improve classifier performance in CHAR.

Thus, in this research, in contrast to previous works that focused on developing novel and more complex network architectures, we evaluated the process of feature engineering in the performance of a known model. We evaluated the performance of a Deep Neural Network (DNN) using time-domain and frequency-domain features extracted from two datasets, ExtraSensory and WISDM. In addition, a study on the influence of the number and type of sensors was performed to determine a relationship between activities and sensor response.

This work presented balanced accuracy rates of 79.5%, 91.6%, 90.0%, and 82.9% for standing, sitting, lying down, and walking activities, respectively, on the ExtraSensory dataset. This represents improvements of 5 p.p., 4.6 p.p., 11.7 p.p., and 1.3 p.p., respectively, in these context activities compared with ExtraSensory App [19]. Moreover, our feature engineering process also improved the results obtained by Fazli et al. [14] by 8.9 p.p. and 39.5 p.p. considering the Standing and Lying Down labels, respectively, and on WISDM, it provided an improvement of 1.7 p.p. in overall balanced accuracy.

This work also demonstrates that the model’s performance increases as the number of sensors increases, showing a direct correlation between the number of sensors and model performance. Additionally, we show that smartphone-specific sensors, such as location and phone state sensors, improve activity recognition.

## 3. Materials and Methods

This section describes the approach and experiments performed using the DNN to study the contribution of feature engineering and feature selection to improving classifier performance. In addition, a comparison is made between ExtraSensory App [9] and our model approach to corroborate the results.

### 3.1. The Datasets

To perform the experiments on feature engineering, we used the ExtraSensory and WISDM datasets, which are employed in many context classifiers and approaches.

#### 3.1.1. ExtraSensory Dataset

The ExtraSensory dataset is public and has raw data available, which makes it suitable for further signal processing, such as FFT, to extract features from signals. Therefore, we combined the signal of different sensors extracted from ExtraSensory raw data to perform feature selection.

ExtraSensory is formed by data collected from 60 people *in the wild*, or else, data that were acquired while people were doing an activity, with no strategy control. In addition, the subjects used Android and IOS systems to collect data, and the participants were of different ages, sex, and ethnicity [19].

This dataset provides data from sensors such as accelerometers, gyroscopes, magnetometers, gravity sensors, compasses, luminosity sensors, audio (microphone) sensors, and battery state sensors. Accelerometer, gyroscope, and magnetometer signals were sampled at 40 Hz each [9].

These sensors may work differently based on the device that is being used. A variety of smartphones were used in this dataset, including iPhones of the 4th generation containing a 3D accelerometer (LIS331DLH) [30] and a 3D gyroscope (L3G200D) [31], both manufactured by ST-Microelectronics based in Geneva, Switzerland.

However, the 3D magnetometer (AKM8975) [32] was manufactured by Asahi Kasei Microdevices based in Nobeoka, Japan. In the 5th- and 6th-generation devices, the 3D magnetometer was updated to a newer model (AK8963) [33] by the same company.

Although the 3D gyroscope stayed the same as that of devices of the previous generation, the 3D accelerometer was completely changed to the BMA220 model by Bosch Sensortec, Gerlingen, Germany [34]. Many different models of Android-based smartphones were used in this dataset, such as Samsung, Nexus, HTC, Moto G, LG, Motorola, One Plus One, and Sony.

Most notably, Samsung Galaxy Nexus is equipped with the same 3D accelerometer used in the latter phone models. However, it includes a 3D gyroscope (MPU-3050) [35] by InvenSens (San Jose, CA, USA) and a 3D magnetometer (YAS530) [36] manufactured by Yamaha in Shizuoka, Japan [37].

All sensors mentioned in this work are in agreement with the IEEE standards of measurements and operation modes [38].

All data are available in short time series, 20 s windows, which resulted in more than 300 K labeled samples in two formats: processed and raw data [9]. The dataset labels encompass a set of 116 labels divided into two groups: primary and secondary labels. The primary set contains posture-related activities and seven multiple exclusive instances. The secondary labels comprise 109 context activities that are not mutually exclusive, i.e., more than one label can coexist [19].

#### 3.1.2. WISDM

We also evaluated our approach on another dataset, WISDM [39]. This dataset only has 2 sensors, an accelerometer and a gyroscope, and 18 activities: walking, jogging, stairs, sitting, standing, kicking (soccer ball), dribbling (basketball), playing catch (tennis ball), typing, writing, clapping, brushing teeth, folding clothes, eating pasta, eating soup, eating a sandwich, eating chips, and drinking.

The data were collected from 51 subjects, who were asked to perform the activities for 3 min. The sensors were sampled at a rate of 20 Hz. We extracted the same features as from ExtraSensory in both time and frequency domains.

### 3.2. Feature Extraction

Intending to perform feature engineering, we selected only raw data from the accelerometer, gyroscope, magnetometer, gravity sensors, and compass sensors. These sensors were chosen because after exploratory analysis and data cleaning, they had more remaining samples to train the models.

Additionally, the data are time-series data, which are more related to the main goal of our paper, i.e., comparing time-series features and frequency-series features. We also performed experiments by adding location (Loc) and phone state (PS) for completeness, as reported in Section 4.2. Table 1 briefly describes the sensors, and the scheme is illustrated in Figure 1.

#### Discrete Fourier Transform

The Discrete Fourier Transform is used to change data’s actual space to data’s frequency space. The Fourier Transform is used in data and signal processing applications. This approach is used to take advantage of computational efficiency in some applications and because of all the preserved signal information.

In order to extract the frequency characteristics of the sensors’ signals, we used the Discrete FFT (DFFT). The sample rate was calculated using a 20 s window and the number of samples obtained for every sensor, as Equation (Equation 1) shows:(1)$$sampleRate=\frac{n_{samples}}{20}.$$

The DFT results in an array corresponding to the Fourier frequency coefficients of the referred time-domain signal window. The features were extracted using statistical measures of the Fourier coefficients. Table 2 illustrates the description of every extracted feature and the equations to calculate them.

### 3.3. Model Description and Training

The model used was based on the approach by Fazli et al. (2020) [14], which uses a hierarchical classifier that recognizes an activity at a higher level, i.e., whether it is stationary or non-stationary. Afterwards, at a lower level, if the high-level activity is classified as stationary, the lower-level activity may be subclassified as Standing, Sitting, or Lying Down; the same as applies to non-stationary classification, whereby activities are subclassified as Running, Bicycling, or Walking.

In our approach, we train five models. The first classifier is used to classify the higher-level labels, called primary labels, and the other four classifiers correspond to the lower-label classifiers. We used the labels Sitting, Standing, Walking, Lying Down, Running, and Bicycling for the primary model.

The secondary model used relies on the result of the primary model; for each label at the primary level, a model is trained to predict second-level classification, e.g., for the label Sitting (primary), a classifier is trained to predict the lower label Surfing the Internet, Watching TV, In a Meeting, or In a Car. Specifically for the primary labels Running and Bicycling, the model infers Exercise for the lower label just for convention. The architecture can be viewed in Figure 2.

The training and evaluation strategy adopted was the stratified K-fold cross-validation with 5 folds because of the imbalanced classes in the dataset. The entire dataset was split into two subsets, one for training (80% of the total of instances) and the second for testing (20% of the total of instances), and standard normalization was used on the data [40]. We also separated 2% of the training set for validation during the training phase, and the best model was saved and used for further evaluation using the remaining test data. Figure 3 illustrates the strategy used for training all models.

Every classifier was trained for 100 epochs with a batch size of 10, with one input layer, one hidden layer, and one output layer. After the first layer, a dropout layer was added, and all layers were fully connected. Table 3 indicates the parameters of the used model. The input layer sizes were 130 for frequency-domain features and 92 for time-domain features, while the output layer size varied according to the number of labels of each model. The metric used for choosing the best model was accuracy.

### 3.4. Experiments

We conducted two types of experiments. The first aimed to assess how the number and combination of sensors affected recognition performance. Furthermore, the second type of experiment aimed to answer whether features extracted from the DFT could be used for context evaluation.

#### 3.4.1. Influence of Sensors

To evaluate the sensors’ impact, we separated the sensors into two groups. One group comprised inertial sensors: accelerometer, gyroscope, magnetometer, compass, and gravity sensor. The other group comprised context sensors, i.e., location and phone state sensors. We chose the model proposed in [14] to achieve the objective and evaluated it by varying the number of, combining, and mixing contextual and inertial sensors.

We compared using one to five sensors and evaluated the model’s overall accuracy. Furthermore, we measured the specific contribution of every sensor and the combination of them using balanced accuracy.

Table 4 shows the number of experiments performed concerning the number of sensors. A description of the combinations, along with the number of labeled instances for every combination, is provided in Table 5, Table 6 and Table 7.

Additionally, we also included context sensors to evaluate the model. We experimented using the five inertial sensors along with the location and phone state sensors.

Furthermore, to maintain the experiment’s randomness, we used a constant seed to maintain the pseudo-shuffling of the data.

The difference in the number of samples in every experiment was due to the unbalanced measurements performed with every sensor. For instance, the accelerometer was the sensor with the highest number of measures, while the compass had the lowest number of samples. The number of samples differed because lines with empty values (NaN) were dropped during data pre-processing.

ExtraSensory has imbalanced labels. For instance, while the number of samples for the label “Running” is less than a thousand, "Sitting" has more than ten thousand samples. When a model sees more of one label, it may become biased and learn more about one class than others. This is dangerous for CHAR, since it leads to applications making wrong decisions or inadequate recommendations. Because of that, the stratified K-fold training strategy was used to train our model.

The total number of models trained and evaluated was 5, 1 to classify five primary labels and 4 to classify context situations as described in Figure 2, and their performances were evaluated using accuracy, balanced accuracy, sensitivity, and specificity as evaluation metrics.

#### 3.4.2. Time-Domain Features vs. Frequency-Domain Features

The hypothesis behind this experiment is that features directly extracted from time series are prone to errors due to artifact movement and interference and carry less information than features extracted from a signal transformed using the DFT. Thus, the same model performs better using DFT-extracted features.

To prove this hypothesis, we used two datasets, Extrasensory and WISDM, to compare the performance of the same architecture using time-extracted features and DFT-extracted features. Time-based features were calculated as in Vaizman et al.’s [9] study on both datasets.

The sensor signals used for the time- and frequency-domain features included accelerometer, gyroscope, magnetometer, and compass signals. This experiment did not include gravity because it lacks compiled features in the original ExtraSensory dataset.

To conduct experiments with WISDM, which only contains time-series data obtained with sensors, we first had to split the 3 min activities into smaller records containing 20 s windows with an overlap between successive windows of 50%. After this process, we calculated the trivial statistics and FFT to compare the same model using feature engineering approaches.

The difference between the two approaches was evaluated using the following metrics: accuracy, balanced accuracy, sensitivity, and specificity.

## 4. Results

This section summarizes the most valuable results found in our experiments involving feature engineering, feature selection, and our model’s final performance compared with models by other authors. It also explains the metrics used in the experiments and their importance. Moreover, this section also presents the results obtained in the exploratory data analysis performed on the ExtraSensory dataset.

### 4.1. Metrics

Since ExtraSensory has imbalanced classes, we evaluated our approach using balanced accuracy (*BA*) as the main metric. In addition, accuracy, sensitivity, and specificity were also used. Accuracy, balanced accuracy, sensitivity, and specificity are calculated using Equations (Equation 2), (Equation 3), (Equation 4), and (Equation 5), respectively.
(2)$$A=\frac{TP+TN}{TP+TN+FP+FN}.$$
(3)$$BA=\frac{sensitivity+specificity}{2}.$$
(4)$$sensitivity=\frac{TP}{TP+FN}.$$
(5)$$specificity=\frac{TN}{TN+FP}.$$
where the following apply:
*TP*: number of true-positive cases;*TN*: number of true-negative cases;*FP*: number of false-positive cases;*FN*: number of false-negative cases.

### 4.2. Sensor Contribution Evaluation

To evaluate the sensors’ contributions to model performance, we evaluated the contribution of the number of sensors, type (inertial or context), and the combinations of different inertial sensors to the performance of our chosen model. Table 8 shows the results of model performance obtained by varying the number and type of sensors.

In most cases, using five sensors led to better results than smaller sensor combinations, except for the Walking and Lying context models. In addition, using context sensors (location and phone state) also led to an increase in performance when compared with only using inertial sensors.

In Table 8, it is also possible to see that there is a decrease in the balanced accuracy of the secondary models in comparison with the primary model. This is a problem of the architecture because of error propagation. A wrong prediction in the primary model consequently leads to a wrong prediction in the secondary model. However, the effect of the number of sensors and the context sensor is remarkable. We had improvements of 11.8%, 5.9%, and 6.3% in the primary, secondary (Sitting), and secondary (Walking) networks, respectively.

We also compared the individual contributions of inertial sensors to classifying different activities. We used the accelerometer, magnetometer, gyroscope, compass, and gravity sensor to classify the six primary activities, Standing, Sitting, Lying Down, Running, Walking, and, Bicycling.

Figure 4 and Figure 5 show the balanced accuracy response of individual sensors and all possible combinations of three sensors, respectively. It is possible to observe that a sensor can contribute more to a specific activity, i.e., the magnetometer performed better in recognizing Lying Down and Walking activities while performing poorly in recognizing Standing.

Comparing the overall performance shown in Figure 4 and Figure 5, it is also possible to observe that with more sensors, we can have better performance.

Since in our hierarchical model we infer the primary label before inferring the secondary one, our model’s accuracy in retrieving each primary label is the most important. Therefore, we calculated the BA and sensitivity for each primary label. Sensitivity is relevant because it is focused on measuring how many of the positive samples have been correctly classified. The results are presented in Table 9.

The sensitivity for the labels Standing and Walking was considerably lower than that for other labels. Therefore, the inference of the secondary labels derived from them was impaired compared with the primary model, leading to 76.3% and 84.5% of failures, respectively, being caused by this.

However, while the label Lying Down had high BA and sensitivity, the secondary labels had poor results. The overall BA of the second model was 50.7 ± 2%, and the sensitivity was 1.7 ± 4%, but only 33.1% of errors came from the primary model. This means that our model has difficulty in distinguishing whether a person lying down is surfing the internet, watching TV, or sleeping, given the five sensors (accelerometer, gyroscope, magnetometer, compass, and gravity sensor).

On the other hand, the secondary model that inferred additional labels for Sitting had 74.9 ± 7% BA and 57.5 ± 13% sensitivity. Therefore, these sensors are enough to help our model learn if a person sitting is surfing the internet, watching TV, in the middle of a meeting, or inside a car. Interestingly, the labels Surfing the Internet and Watching TV were used for Lying Down and Sitting, but the inference of the latter was much better. It may have been because Sitting had more positive samples than Lying Down.

Although the BA rates from the experiments on Standing, Walking, and Lying Down were 54 ± 2%, 51.6 ± 6%, and 50.7 ± 2%, this was primarily due to the high specificity. The sensibility was very low, ranging from 0% to 13%. While, in part, this is because our dataset was imbalanced and lacked positive samples, the fact that we discarded samples that were incorrectly classified in the primary model significantly impacted the decrease in sensibility.

Hence, using the primary label to infer the secondary one without relying entirely on its accuracy is desirable. For instance, the primary label may be used as an attribute of the sample. During training, we can augment the data by including samples of the secondary label with an incorrect primary label to help the model correctly predict a secondary label, even if the primary one is incorrect.

#### Context Recognition

Besides the mentioned primary labels, the ExtraSensory dataset has many contextual labels. We selected eight context labels based on the available data: Talking, Sleeping, Eating, Watching TV, Surfing the Internet, With Friends, Computer Work, and With Co-Workers. The inference results using five sensors are in Table 10. Overall, the BA was 88.9%. Thus, the model can infer details about users’ routines.

### 4.3. Comparison between Time Features and DFT-Transformed Features

The authors of the ExtraSensory dataset published the results of a technique using only one NN to predict 51 labels. Thus, we used it as a baseline to compare our results. We used the same samples from the experiments with our technique and the trained model *es5sensors*, which is available at the repository of ExtraSensory App (https://github.com/cal-ucsd/ExtraSensoryAndroid, accessed on 8 June 2023) [41].

Since the NN is already trained, the inputs are fixed. Hence, we used the same sensors they used: accelerometer, gyroscope, location, quick location features, audio naive, and discrete sensors. In addition, the sensors obtained from a temporal series were in the time domain. Table 11 presents the results.

The primary model HHAR for both time and frequency domains obtained a better result than ExtraSensory App. Since ExtraSensory App was trained to classify samples into more classes (51) and the HHAR model only classifies our 6 primary labels, it was easier for them to learn how to distinguish the samples of these labels.

However, the secondary models had poor results compared with ExtraSensory App. It could have been due to samples failing at the primary model application and being considered FN. We also conducted experiments predicting samples directly on the secondary models to verify this hypothesis. In this way, we assumed that the samples were correctly classified at the primary level. The comparison is in Table 12.

Although all of the frequency models performed better than the experiment relying on the inference of the primary model, the time model still had hindrances in some cases. For instance, the overall BA of the Lying Down model using time features was low because of the specificity of Sleeping, which was 4.2%, and the sensibility rates of Watching TV and Surfing the Internet, which were 0.5% and 7.2%, respectively.

Furthermore, the frequency feature models had a higher BA than ExtraSensory App. This was achieved using fewer sensors than ExtraSensory App, since we only used the five sensors that can be extracted using the DFFT in our technique.

To investigate the generalization of our method, we applied the same feature engineering method to WISDM. We compared it with the features extracted from the time series of sensors using balanced accuracy. The average BA predicting all 18 labels is in Table 13. It demonstrates that using frequency features performed better than using time features on WISDM. Our technique also generalizes to diverse activities since it recognizes all WISDM labels.

## 5. Discussion

This section presents a discussion about the main results and comparisons, presenting the answers to the central questions of this work:
1How can the number of sensors influence model performance in CHAR tasks?2Does a sensor contribute differently to different activities?3Are DFT-extracted features more suitable than trivial statistic features?

The results shown in Table 8 indicate that the increase in the number of sensors directly impacted the increase in model performance in most cases. Moreover, using context-related sensors is also essential in the task of CHAR, because they can improve performance with no need to use more complex algorithms that demand more computational resources.

Another conclusion drawn from this study is that choosing suitable sensors is essential for application, because different sensors contribute to different activities. Figure 4 and Figure 5 show these differences with respect to six activities. This is important because it can be used in applications with limited computational or hardware resources.

We used two datasets to apply feature engineering using DFT features. We compared them with time-extracted features using the same Neural Network model to prove that DFT-extracted features are more suitable and can enhance model performance using different data. This result can be seen in Table 12 and Table 13. These results show an improvement of 1.7 p.p. using the WISDM dataset and general improvements in the secondary models Standing, Sitting, Lying Down, and Walking by 8.9 p.p., 0.2 p.p., 39.5 p.p., and 0.4 p.p., respectively.

Using sensor information instead of cameras for CHAR tasks is also good in terms of privacy concerns, as data can be encrypted at the user level and at the application level, as in [42]. Moreover, the device containing all necessary sensors can be carried by the user all the time, unlike cameras.

Our tests employed a simple Neural Network design. However, we believe that the feature engineering module has the capability to significantly improve cutting-edge approaches such as Ge et al.’s [43].

Finally, although a more complex classifier can perform better, depending on the application and system architecture, it costs more due to requiring more time in the training and fine-tuning processes. This may lead to a longer time to market due to the prototyping process. Moreover, the inference process can also be costly, requiring more hardware and costing more money to the final user.

## 6. Conclusions

The process of feature engineering is very important in any ML modeling, as data carry important information. In CHAR tasks, data come from sensors. These signals, in most cases, are combined with noise, which can be caused by user movements and electromagnetic interference in the device circuit.

Thus, choosing the right features is an important task. In this work, not only we studied the difference between the two ways of feature extraction, but we also performed a study on the influence of the number and type of sensors that can improve ML or Deep Learning model performance based on the Extrasensory and WISDM datasets.

The experiments and results described in Section 3.4.1 and Section 4.2, respectively, proved that the number of sensors used in the input of the classifier and classifier performance have a positive correlation. This happens because each sensor signal has a correlation with the activity characteristics, similar to an observer seeing the same scene from different angles.

The type of sensor is also important. Our work shows that a specific sensor, such as an accelerometer, contributes in different ways to distinct activities and distinct contexts. Thus, mixing different types of inertial sensors and different information from context sensors, such as location, may improve the performance of CHAR applications without using more complex classifiers.

To achieve good performance in CHAR applications, using as many sensors as possible is not the only path. Using the correct information is also important. After studying the difference between time-extracted and DFT-extracted features, we conclude that extracting features from DFT signals performs better given the same ML algorithm and the same set of sensors.

The findings of this study can be applied in many applications in the area of CHAR, as by using the right set of sensors and DFT features, we can improve a CHAR application that runs on wearable devices. Normally, these devices are equipped with less memory space and less computational power and rarely have all kinds of sensors available to obtain users’ information in real time.

Finally, as future work, the correlation analysis between the type of sensor and activities can be expanded to more activities and contexts in order to explore more applications, such as sleep-monitoring routine recommendation systems. Another approach is to evaluate a user’s routine diary and how it may affect sleep.

## Figures and Tables

**Figure 1 sensors-23-05756-f001:**
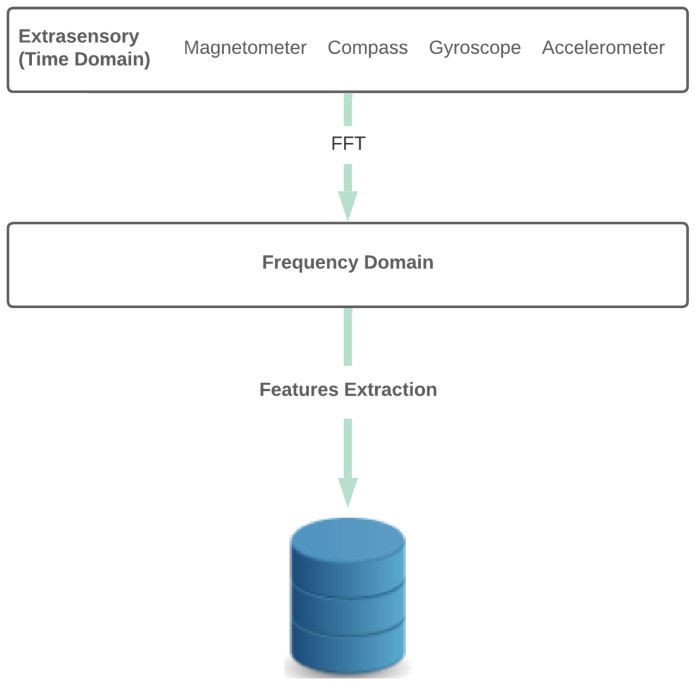
Feature extraction process. The DFT was used on the sensors’ signals, and the frequency coefficients were used to extract the features.

**Figure 2 sensors-23-05756-f002:**
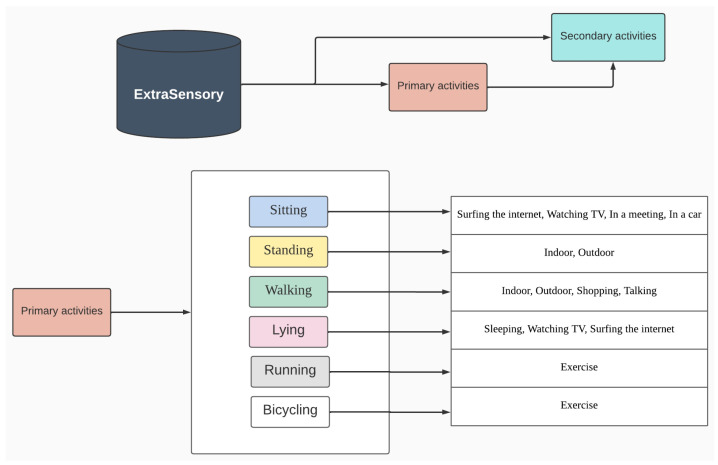
Model architecture. The first model infers the primary activity, and according to it, we use a second model that infers the context related to that activity.

**Figure 3 sensors-23-05756-f003:**
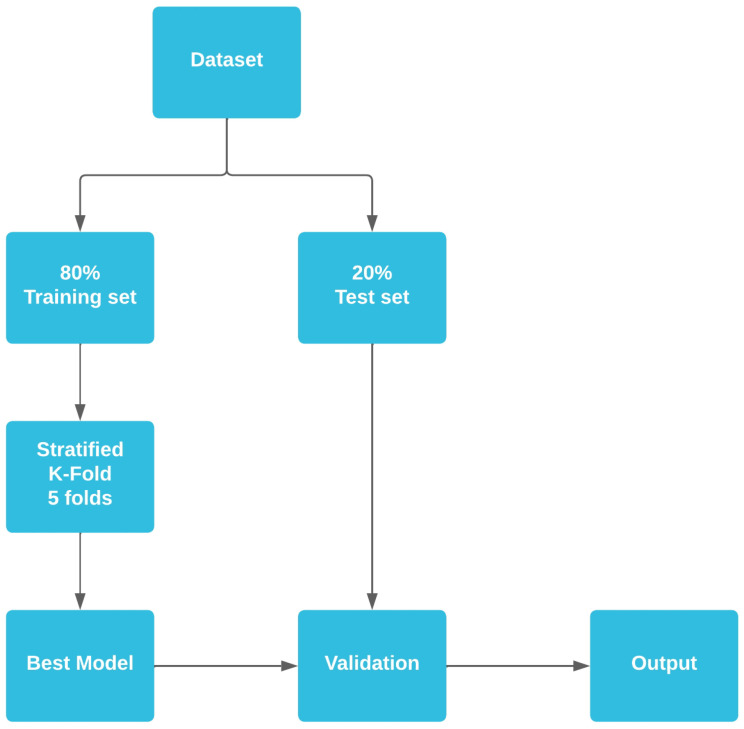
Strategy used on frequency-domain dataset to train the models. The data were partitioned into training and test splits, and the model was evaluated using the test partition.

**Figure 4 sensors-23-05756-f004:**
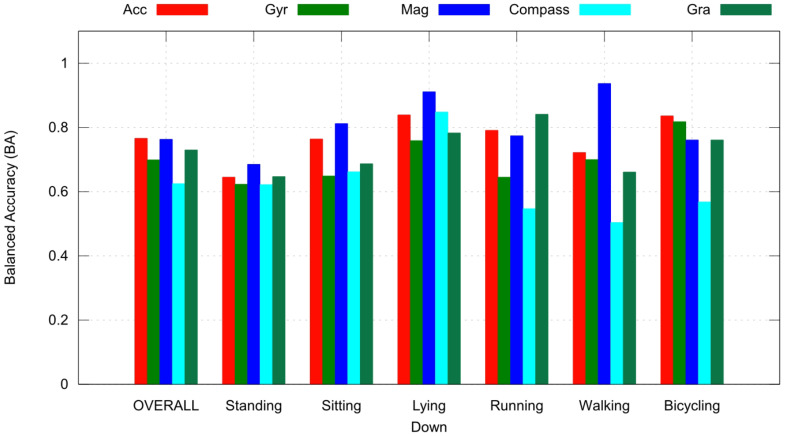
BA for each experiment and each label, and overall performance (the mean between label results) of the primary model.

**Figure 5 sensors-23-05756-f005:**
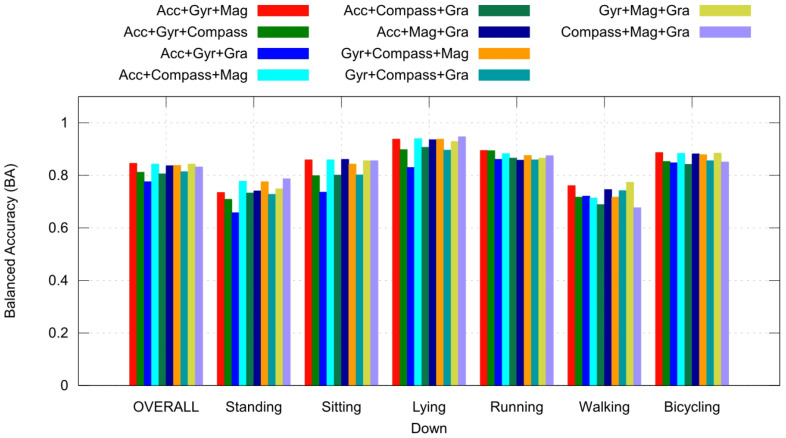
BA for experiments with 3 sensors for each primary label and overall performance (the mean between label results).

**Table 1 sensors-23-05756-t001:** Description of sensors used and measurements.

Sensor	Description
Accelerometer	Tri-axial direction and magnitude of acceleration.
Gyroscope	Rate of rotation around phone’s three axes.
Magnetometer	Tri-axial direction and magnitude of the magnetic field.
Smartwatch compass	Watch heading (degrees).
Gravity	Estimated gravity.
Loc	Latitude, longitude, altitude, speed, accuracies, and quick location variability.
PS	App state, battery state, WiFi availability, on the phone, time of day.

**Table 2 sensors-23-05756-t002:** Equations used for feature extraction.

Feature	Description	Equation
AV	Coefficient average	$AV=\frac{1}{N}\sum_{k=1}^{N}F\left[k\right]$
F[0]	Signal DC value	F[0]
SD	Standard deviation	$SD=\sqrt{\frac{\sum_{k=1}^{N}{\left(F\left[k\right]-\mu \right)}^{2}}{N-1}}$
MIN	Minimum value	$MIN=min_{N}F[k]$
MAX	Maximum value	$MAX=max_{N}F[k]$
RG	Range	RG=max−μ
FQ	First quartile	$FQ=F[\frac{N+1}{4}]$ term
SQ	Second quartile	$SQ=F[\frac{N+1}{2}]$
RMS	RMS value	RMS = $\sqrt{\frac{\sum_{k=1}^{N}{\left||F\left[k\right]|\right|}^{2}}{N}}$
MED	Median	$med=F[\frac{N+1}{2}]$ for N even $med=\frac{F\left[\frac{N+1}{2}\right]+F\left[\frac{N-1}{2}\right]}{2}$ for N odd

**Table 3 sensors-23-05756-t003:** Model parameters.

Parameter	Value	Activation Function
Max Input Layer	130 (frequency) 92 (time)	ReLU
Hidden layer 1	64 neurons	ReLU
Dropout layer	0.3	
Hidden layer 2	128 neurons	ReLU
Output layer	No. of labels	Softmax
Optimizer	Adam	
Learning rate	0.001	
Loss function	Sparce Categorical Cross-Entropy	
Batch size	10	

**Table 4 sensors-23-05756-t004:** Total combination of experimental scenarios per model.

Number of Sensors	Number of Experiments
1	5
2	10
3	10
4	5
5	1
**Total**	31

**Table 5 sensors-23-05756-t005:** Number of samples for experiments with 1 sensor for each primary label.

Sensor	Samples	Lying Down	Sitting	Walking	Running	Bicycling	Standing
Acc	51,159	18,174	19,353	3173	723	2421	7315
Gyr	48,936	17,597	18,407	2986	711	2374	6861
Mag	46,152	17,044	17,467	2592	635	2082	6332
Compass	24,104	4477	11,038	1448	405	1606	5130
Gra	48,936	17,597	18,407	2986	711	2374	6861

**Table 6 sensors-23-05756-t006:** Number of samples for experiments with 2 sensors for each primary label.

Sensors	Samples	Lying Down	Sitting	Walking	Running	Bicycling	Standing
Acc + Gyr	48,936	17,597	18,407	2986	711	2374	6861
Acc + Mag	46,152	17,044	17,467	2592	635	2082	6332
Acc + Compass	24,104	4477	11,038	1448	405	1606	5130
Acc + Gra	48,936	17,597	18,407	2986	711	2374	6861
Mag + Gyr	46,152	17,044	17,467	2592	635	2082	6332
Mag + Gra	46,152	17,044	17,467	2592	635	2082	6332
Mag + Compass	21,724	4231	10,089	1190	370	1372	4472
Gyr + Gra	48,936	17,597	18,407	2986	711	2374	6861
Gyr + Compass	23,372	4413	10,661	1417	404	1601	4876
Gra + Compass	23,372	4413	10,661	1417	404	1601	4876

**Table 7 sensors-23-05756-t007:** Number of samples for experiments with 3 to 5 sensors for each primary label.

Sensors	Samples	Lying Down	Sitting	Walking	Running	Bicycling	Standing
Acc + Gyr + Mag	46,152	17,044	17,467	2592	635	2082	6332
Acc + Gyr + Compass	23,372	4413	10,661	1417	404	1601	4876
Acc + Gyr + Gra	48,936	17,597	18,407	2986	711	2374	6861
Acc + Mag + Compass	21,724	4231	10,089	1190	370	1372	4472
Acc + Gra + Compass	23,372	4413	10,661	1417	404	1601	4876
Acc + Mag + Gra	46,152	17,044	17,467	2592	635	2082	6332
Gyr + Mag + Compass	21,724	4231	10,089	1190	370	1372	4472
Gyr + Gra + Compass	23,372	4413	10,661	1417	404	1601	4876
Gyr + Mag + Gra	46,152	17,044	17,467	2592	635	2082	6332
Mag + Gra + Compass	21,724	4231	10,089	1190	370	1372	4472
Acc + Gyr + Mag + Compass	21,724	4231	10,089	1190	370	1372	4472
Acc + Gyr + Mag + Gra	46,152	17,044	17,467	2592	635	2082	6332
Acc + Gyr + Gra + Compass	23,372	4413	10,661	1417	404	1601	4876
Acc + Mag + Gra + Compass	21,724	4231	10,089	1190	370	1372	4472
Gyr + Mag + Gra + Compass	21,724	4231	10,089	1190	370	1372	4472
Acc + Gyr + Mag + Gra + Compass	21,724	4231	10089	1190	370	1372	4472

**Table 8 sensors-23-05756-t008:** Overall BA combining the accelerometer, gyroscope, magnetometer, compass, and gravity sensor. Additionally, we evaluated these 5 sensors with location and phone state. We summarize the average results per quantity of sensors.

No. of Sensors	Primary Network	Secondary (Sitting)	Secondary (Standing)	Secondary (Walking)	Secondary (Lying)
1 sensor	73.0%	66.6%	51.2%	50.9%	50.0%
2 sensors	77.8%	73.2%	54.0%	51.9%	50.4%
3 sensors	83.4%	78.5%	54.0%	52.1%	51.9%
4 sensors	84.2%	78.9%	56.3%	51.8%	51.3%
5 sensors	84.6%	77.7%	57.9%	54.6%	58.2%
5 sensors + Loc and PS	96.4%	83.6%	64.2%	54.1%	57.3%

**Table 9 sensors-23-05756-t009:** BA and sensitivity for each primary label.

Label	BA (Mean ± SD)	Sensitivity (Mean ± SD)
Standing	72.8 ± 5%	51.3 ± 13%
Sitting	81.2 ± 6%	85.4 ± 12%
Lying Down	91.1 ± 5%	89.0 ± 4%
Running	86.1 ± 7%	72.3 ± 16%
Walking	71.7 ± 7%	45.7 ± 11%
Bicycling	85.1 ± 6%	71.3 ± 12%

**Table 10 sensors-23-05756-t010:** BA of context labels recognition in ExtraSensory dataset.

Label	BA
Talking	97.0%
Sleeping	95.1%
Eating	79.8%
Watching TV	91.8%
Surfing the Internet	92.6%
With Friends	73.4%
Computer Work	96.6%
With Co-Workers	84.7%
**OVERALL**	**88.9%**

**Table 11 sensors-23-05756-t011:** Comparison of BA between our technique and ExtraSensory App inferring primary and secondary labels.

Technique	Primary Network	Secondary (Standing)	Secondary (Sitting)	Secondary (Lying Down)	Secondary (Walking)
ExtraSensory App	76.0%	74.5%	87.0%	78.3%	81.6%
HHAR (time)	87.1%	57.3%	80.6%	52.7%	55.5%
HHAR (frequency)	85.1%	58.4%	72.8%	50.1%	50.8%

**Table 12 sensors-23-05756-t012:** Comparison of secondary model’s BA between our technique without using primary model prediction and ExtraSensory App.

Technique	Secondary (Standing)	Secondary (Sitting)	Secondary (Lying Down)	Secondary (Walking)
ExtraSensory App	74.5%	87.0%	78.3%	81.6%
HHAR (time)	70.6%	91.4%	50.5%	82.5%
HHAR (frequency)	79.5%	91.6%	90.0%	82.9%

**Table 13 sensors-23-05756-t013:** Comparison of BA between time-domain and frequency-domain features in WISDM.

Technique	BA
HHAR (time)	86.7%
HHAR (frequency)	88.4%

## Data Availability

Suggested Data Availability Statements are available in Section 4.3.

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
