# Peer review of "A Study on the Influence of Sensors in Frequency and Time Domains on Context Recognition"

_sensors, 2023, doi:10.3390/s23125756_

Round 1

Reviewer 1 Report

This paper studies the influence of time- and frequency-domain data/sensors on context recognition. This aims to highlight the importance of using the correct features and the correct sensors to achieve a better performance with less computational cost. I believe that this is a very useful thing to study, especially with most people using random algorithm with random feature selection without thinking much about why they should use one over the other and what side effects that would entail.

The authors made conclusions based on just one dataset, which makes their conclusions limited to this dataset and maybe very similar one. They need to include a bigger diversity and statistically show that their results can generalize.

Authors only used limited sensor types/data which are mainly linked to movement/direction sensors that could be integrated in one wearable sensor node.

The activities analyzed are also limited to basic simple activities, so no conclusions could be drawn with respect to other contexts that need recognition of more sophisticated activities, such as physical therapy activities, eating episodes, etc.

The results, specially in tables 4 and 5, are a little surprising since many existing models can already achieve significantly better performance with the same or a similar set of activities, which suggests that the authors did not use state-of-the-art algorithm in their experiments.

Overall, this is a good start, but the experimental procedure needs to significantly improved for the results to be useful to other researchers in the field.

The English of this paper definitely needs revision.

Author Response

Thank you for your contribution. Please see the attachment.

Reviewer 2 Report

The paper reports a study of the influence of sensors in frequency and time

domain for context recognition. All Extrasensory datasets were collected from the database of literature, and not measured by the authors. At this stage, the style of this paper was like machine learning software using games, not a research paper. It did not have a significant contribution to the field of Sensors or Measurement. The paper is suited to submission to related journals about machine learning.

1.     What was the standard to separate data into 80% training and 20% test dataset? Why not use the 50% training and 50% test dataset? That is the key to ensuring the correctness of the model.

2.     What was the effect of the sensors’ performance on the results of machine learning?

Please make the connection between the sensors’ performance and the validation of the machine-learning model. That is, let the content of this paper fit the requirement of the scope of the Sensor journal. Not only use the data sets of literature and commercial software.

Please check the spelling of words carefully.

Author Response

(The authors gave the same response as above.)

Reviewer 3 Report

this paper aims to study the influence of sensors in frequency and time domain for context recognition.

Overall, I think the work of this paper is important and meaningful. The paper is well written and structured. Besides, the origination structure of this paper is complete, which includes the presentations of theory itself, implementation, and experimental evaluation and result analysis. Therefore, I think this paper can be accepted after some minor revisions.

(1) Method. 1) The readability of this paper may be improved by adequately explaining the symbols, terms and concepts used in the paper, providing sufficient background on the investigated topic, as well as providing some animated examples. 2) This paper directly describes definitions and preliminaries, without introducing the background information. 

3) The language presentation also needs to be improved. Some sentences are too long to be understood. 

(2) Conclusion. There are no real insightful conclusions drawn from the study and no suggestions for practical use of the results. Therefore, the conclusion section should be totally rewritten in order to 1) more clearly highlight the theoretical and practical implications of the research; 2) Indicate practical advantages; 3) Discuss research limitations; and 4) supply 2-3 solid and insightful future research suggestions.

(3) Discussion. The article contains a lot of citations, but some relevant literatures are still missing. Also, I suggest to add a wider discussion on the impact of the proposed method on related domains such as privacy. Here are some suggestions for authors to go through and follow the direction for reworking. e.g., a1) A Dummy-based User Privacy Protection Approach for Text Information Retrieval; a2) How to Ensure the Confidentiality of Electronic Medical Records on the Cloud: A Technical Perspective; a3) A Basic Framework for Privacy Protection in Personalized Information Retrieval; a4) Constructing Dummy Query Sequences to Protect Location Privacy and Query Privacy in Location-Based Services; a5) An Effective Approach for the Protection of User Commodity Viewing Privacy in E-commerce Website; a6) A Confusion Method for the Protection of User Topic Privacy in Chinese Keyword Based Book Retrieval; a7)  Optimal Privacy Preservation Strategies with Signaling Q-Learning for Edge-Computing-Based IoT Resource Grant Systems

(4) I suggest the authors make a comprehensive investigation on the medical processing methods in the literature in the introduction part and give the analysis to the existing works such as (https://doi.org/10.1016/j.compbiomed.2022.105918,https://doi.org/10.1016/j.compbiomed.2022.106297,https://doi.org/10.1016/j.compbiomed.2022.105326,https://doi.org/10.1016/j.compbiomed.2022.106300,https://doi.org/10.1016/j.compbiomed.2022.105929) to make the whole work more in-depth.

Moderate editing of English language

Author Response

(The authors gave the same response as above.)

Round 2

Reviewer 2 Report

All problems replied adequately.

Minor editing of the English language required